# Machine learning approach for automatic recognition of tomato-pollinating bees based on their buzzing-sounds

**Alison Pereira Ribeiro[1], Nádia Felix Felipe da Silva[1], Fernanda Neiva Mesquita[1], Priscila de Cássia Souza Araújo[2], Thierson Couto Rosa[1], José Neiva Mesquita-Neto[3]***

**1** Instituto de Informática, Universidade Federal de Goiás, Goiánia, Goiás, Brazil, **2** Programa de Pós-graduação em Zoologia, Universidade Federal de Minas Gerais, Belo Horizonte, Minas Gerais, Brazil, **3** Centro de Investigación en Estudios Avanzados del Maule, Vicerrectoría de Investigación y Postgrado, Universidad Católica del Maule, Talca, Chile

* jmesquita@ucm.cl

**Data Availability Statement:** The dataset and software codes are available in the GitHub

## Abstract

Bee-mediated pollination greatly increases the size and weight of tomato fruits. Therefore, distinguishing between the local set of bees–those that are efficient pollinators–is essential to improve the economic returns for farmers. To achieve this, it is important to know the identity of the visiting bees. Nevertheless, the traditional taxonomic identification of bees is not an easy task, requiring the participation of experts and the use of specialized equipment. Due to these limitations, the development and implementation of new technologies for the automatic recognition of bees become relevant. Hence, we aim to verify the capacity of Machine Learning (ML) algorithms in recognizing the taxonomic identity of visiting bees to tomato flowers based on the characteristics of their buzzing sounds. We compared the performance of the ML algorithms combined with the Mel Frequency Cepstral Coefficients (MFCC) and with classifications based solely on the fundamental frequency, leading to a direct comparison between the two approaches. In fact, some classifiers powered by the MFCC–especially the SVM–achieved better performance compared to the randomized and sound frequency-based trials. Moreover, the buzzing sounds produced during sonication were more relevant for the taxonomic recognition of bee species than analysis based on flight sounds alone. On the other hand, the ML classifiers performed better in recognizing bees genera based on flight sounds. Despite that, the maximum accuracy obtained here (73.39% by SVM) is still low compared to ML standards. Further studies analyzing larger recording samples, and applying unsupervised learning systems may yield better classification performance. Therefore, ML techniques could be used to automate the taxonomic recognition of flower-visiting bees of the cultivated tomato and other buzz-pollinated crops. This would be an interesting option for farmers and other professionals who have no experience in bee taxonomy but are interested in improving crop yields by increasing pollination.

repository (https://github.com/alisonrib17/bees-tomato).

**Funding:** This work was supported by the Agencia Nacional de Investigación y Desarrollo de Chile (ANID/CONICYT/FONDECYT Iniciación) Research Grant 11190013 to JNM-N (https://www.anid.cl/). The funders had no role in study design, data collection and analysis, decision to publish, or preparation of the manuscript.

**Competing interests:** The authors have declared that no competing interests exist.

## Author summary

Bees are the most important pollinators of cultivated tomatoes. We also know that the distinct species of bees have different performances as pollinators, and these performances are directly related to the size and weight of the fruits. Moreover, the characteristics of the buzzing sounds tend to vary between the bee species. However, the buzzing sounds are complex and can widely vary over time, making the analysis of this data difficult using the usual statistical methods in Ecology. In the face of this problem, we proposed to automatically recognize pollinating bees of tomato flowers based on their buzzing sounds using Machine Learning (ML) tools. In fact, we found that the ML algorithms are capable of recognizing bees just based on their buzzing sounds. This could lead to automating the recognition of flower-visiting bees of the cultivated tomato, which would be a nice option for farmers and other professionals who have no experience in bee taxonomy but are interested in improving crop yields. On the other hand, this encourages the farmer to adopt sustainable agricultural practices for the conservation of native tomato pollinators. To achieve this goal, the next step is to develop applications compatible with smartphones capable of recognizing bees by their buzzing sounds.

## Introduction

Tomato (*Solanum lycopersicum* L.) is the second most important vegetable crop in the world [1]. Global tomato production was around 180, 766 tonnes in 2019 and the production grew 14.1% over the past decade [1]. Despite cultivated tomato being self-pollinated, bee-mediated pollination greatly enhances the quantity and quality of the fruits (greater size and weight), also, contributing to the increase of overall crop productivity [2–9]. The tomato pollinator-dependency is so evident that when it is cultivated in greenhouses, it typically needs to be done by bumblebees that are reared particularly for this purpose, generating an extra cost to the growers [10]. For instance, the pollination service in the tomato crop is estimated at about US$ 992 million/year in Brazil [5].

The morphological specialization of tomato flowers, characterized by the presence of poricidal anthers, restricts the exit of the pollen to a tiny opening sited at the apex of the anther [9, 11, 12]. During visits to these flowers, pollen-collecting bees firmly grasp the anthers and quickly contract their flight muscles, but without moving the wings, producing an audible sound [13, 14]. The resulting vibrations are transferred to the anthers, which shake the pollen inside them, stimulating it to leave by the pores, a phenomenon known as floral sonication or buzz-pollination [12, 14, 15].

Although sonicating bees are among the best pollinators of tomatoes, bees belonging to different taxonomic groups tend to differ in their performance as pollinators [4, 6–9, 16]. In this context, the taxonomic recognition is an indispensable requirement to distinguish among the local set of flower-visiting bees those that are the most efficient pollinators.

However, the huge number of bee species and other insects is a challenge for taxonomists. It is estimated that there are about 20, 000 bee species worldwide [17], and 58% of them, about 11, 600 species of 74 genera, are able to vibrate flowers to extract pollen [18]. Furthermore, the taxonomic identification normally depends on visible morphological characteristics of tiny size, which requires the active participation of experts in the decision-making process, since, for an untrained eye, the species are very similar [19]. Besides that, the decreasing number of taxonomists seriously affects the efficiency of species recognition [20]. This is especially evident in regions where the bee diversity is poorly sampled and underestimated like in Africa,

Asia, and some tropical regions [17]. Therefore, the development and implementation of new technologies that also fulfill taxonomic requirements are needed [20–22].

Due to the limitations of the traditional taxonomy, the automatic classification based on artificial intelligence algorithms has been applied for the identification of plants and animals during the last decades. The automatic classifications based on the recognition of images and/or sounds has been implemented [23–26]. However, recognition based on images is difficult due to complications derived from the orientation of the object, the image quality, the condition of the light, and/or the image background [20]. On the other hand, the sound is relatively easy to acquire and can, in principle, be picked up remotely and continuously [19].

The classification based on Machine Learning (ML) algorithms have demonstrated high efficiency and accuracy for the recognition of animal vocalizations, such as birds and frogs [27–29]. The ML algorithms powered by a method for sound feature extraction (e.g., Mel-Frequency Cepstral Coefficients, Hilbert–Huang Transform) have been also employed for beehives monitoring using audio as one of the inputs (see S1 Table for a detailed description; [30] and references therein). These studies sought to differentiate the bee buzzing sounds from other sounds (cricket chirping and ambient noise) [31], recognize the presence of the queen in a beehive and detect an orphaned colony [32, 33], or identify the circadian rhythm of a honeybee colony [34]. However, only three studies address the problem of automatic bee species classification, and these deal with twelve, two and four classes respectively [19, 35, 36]. Random Forest, Support Vector Machines, and Logistic Regression are the most applied classifiers, and Mel Frequency Cepstral Coefficients (MFCC) is the most used feature extraction strategy (see S1 Table). Although preliminary restricted to a few bees taxa, these studies indicated that the ML algorithms could generate classifiers able to quickly and accurately recognize bee identity.

In this context, the automatic recognition of bees would be especially relevant for the pollination of commercial tomatoes, which need the local native pollinators to enhance the crop productivity [4, 6–8]. Moreover, the professionals typically involved with the management of tomato crops (e.g. farmers, agronomists) have no experience in bee taxonomy. Based on this, we aim to verify the capacity of ML algorithms to automatically recognize the taxonomic identity of visiting bees of tomato flowers based on the characteristics of their buzzing-sounds. In addition, we compared the performance of the ML algorithms and MFCC feature extraction method with classifications based on fundamental frequency realized on the same data set, thus, providing a direct comparison between the two approaches. Due to the high efficiency and accuracy demonstrated by ML tools powered by MFCC features in automatic sound classification, we expected that the join of these two methods would obtain a greater performance compared to classifications based solely on fundamental frequency (hypothesis 1). Additionally, we related the performance of ML algorithms in recognizing bees taxa from buzzing-sounds produced during two different behavioral contexts: flight and sonication. While the flight sound generally has few oscillations and is roughly time-independent [37, 38], the sonication produces more complex sounds that may be associated with intrinsic characteristics of the bee [39–42]. Based on this, we predicted that the buzzing-sounds produced by floral vibrations are more relevant for recognizing bees taxa (hypothesis 2). Therefore, the main contributions of our work are: (1) To evidence the ML classifiers performance in recognizing flower visiting-bees species in relation to the statistical approach used by [43]; (2) To classify a higher diversity of taxonomic groups than previous studies, grouping the largest number of genera and families of bees; (3) To provide evidence of which buzzing sounds is more relevant for taxonomic recognition of the bees by the ML classifiers; (4) To indicate the best ML algorithms that could lead to automatize the taxonomic recognition of flower visiting bees of tomato crops.

## Materials and methods

### Buzzing sounds acquisition

The acoustic recording of buzzes was carried out using tomato plants (*Solanum lycopersicum* ac. BGH 7488) grown at the experimental fields of the Federal University of Viçosa (Minas Gerais, Brazil). A portable hand recorder (SongMeter SM2, Wildlife Acoustics, USA) was used to record the buzzing sounds of bees visiting the tomato flowers. To record the buzzing sounds, a researcher constantly walking through the rows of tomatoes handing-held the recorder while searching the flower-visiting-bees. When the researcher spotted a visiting bee, she carefully approached holding the recorder microphone as close as about 10 cm from the flower being visited. The microphone was constantly pointed toward the bee body, and whenever possible to the dorsum. Sound recordings were obtained for 15 bee species from eight genera and two families (See Table 1). Just after leaving the flower, the bees were captured with an entomological net and placed in glass vials with ethyl acetate, for taxonomic identification. When the researcher was not able to capture a bee individual, the corresponding audio sample was not considered for our analysis. We adopted this procedure for ensuring a correct bee taxonomic recognition, then, the number of bee individuals sampled corresponds to the number of audio files (see Table 1). All bee individuals sampled were identified at the species level by an expert in bee taxonomy.

### Acoustic pre-processing

The original sound recordings (.wav files) were manually classified into two behavioral contexts: (i) Sonication; (ii) flight, see Fig 1. We categorized as *sonication* all the segments of buzzing-sounds produced by bees vibrating tomato flowers and as *flight* the sounds produced by the flying displacement of the bees between tomato flowers, as illustrated in Fig 2. As a result, the set of 59 recordings generated 321 segments, 218 of sonication and 103 of flight (see Table 1). The flight and sonication buzz present pronounced differences in acoustic characteristics, so they can be easily distinguished from the recordings afterward by an experienced

**Table 1. Taxonomic diversity of sonicating bees recorded visiting tomato flowers and the corresponding higher taxonomic group (according to [44]).** (N recordings) denotes the number of individuals with buzzing-sounds recorded; (AF) average frequency ± standard deviation; (Flight segments) the total number of flight segments per species; (Sonication segments) the total number of sonication segments per species.

| Subfamily | Tribe | Genus | Species | AF(±SD) [43] | N recordings | Flight segments | Sonication segments |
|---|---|---|---|---|---|---|---|
| Halictinae | Augochlorini | *Augochloropsis* | *Augochloropsis brachycephala* | 252.1 (±44.8) | 2 | 1 | 3 |
| | | | *Augochloropsis* sp.1 | 203.3 (±17.0) | 5 | 9 | 11 |
| | | | *Augochloropsis* sp.2 | 190.1 (±6.3) | 2 | 2 | 12 |
| | | *Pseudaugochlora* | *Pseudaugochlora graminea* | 210.5 (±10.5) | 3 | 2 | 22 |
| Apinae | Bombini | *Bombus* | *Bombus morio* | 218.3 (±18.9) | 8 | 9 | 19 |
| | | | *Bombus atratus* | 233.9 (±21.7) | 5 | 11 | 15 |
| | Centridini | *Centris* | *Centris tarsata* | 330.8 (±7.3) | 1 | 6 | 2 |
| | | | *Centris trigonoides* | 319.9 (±16.8) | 2 | 8 | 3 |
| | Euglossini | *Eulaema* | *Eulaema nigrita* | 196.7 (±3.5) | 1 | 2 | 2 |
| | Exomalopsini | *Exomalopsis* | *Exomalopsis analis* | 167.6 (±21.8) | 10 | 11 | 63 |
| | | | *Exomalospsis minor* | 151.9 (±12.8) | 4 | 4 | 19 |
| | Meliponini | *Melipona* | *Melipona bicolor* | 337.1 (±29.8) | 8 | 17 | 30 |
| | | | *Melipona quadrifasciata* | 323.0 (±20.9) | 4 | 9 | 8 |
| | Xylocopini | *Xylocopa* | *Xylocopa nigrocincta* | 247.2 (±15.2) | 2 | 9 | 7 |
| | | | *Xylocopa suspecta* | 250.1 (±29.9) | 2 | 3 | 2 |

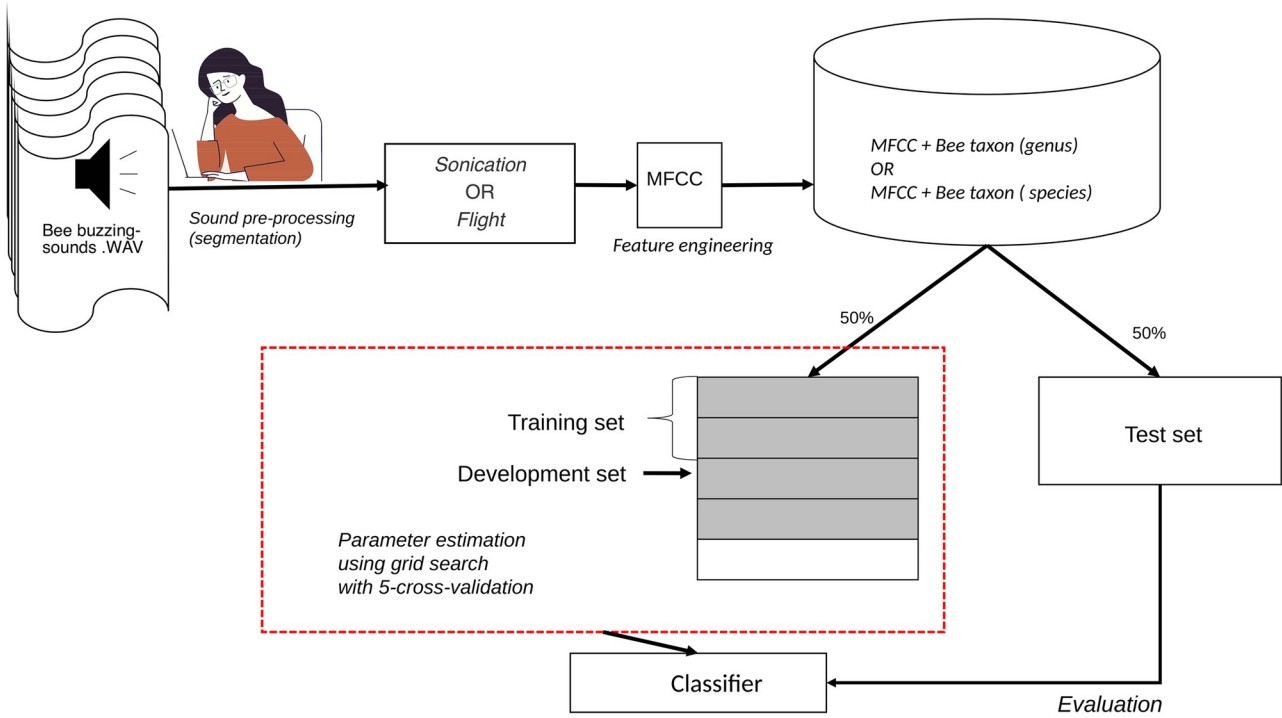

**Fig 1. Overview of the approach adopted for the acoustic classification of bees buzzing-sounds and machine learning workflow.** The original audio files (.wav format) containing recordings of bees buzzing-sounds during visits to tomato flowers were manually classified into sonication or flight segments. Then, the Mel Frequency Cepstral Coefficients method (MFCC) was used to extract the audio features. After, the resulting data set was split into 50% for the training/development set (delimited by the red dashed line) and 50% for the testing data set. The GridSearchCV method was used to tune the hyperparameters of the training set (using 5-cross validations). The test data set was used to evaluate the performance of the Machine-Learning classifiers in correctly assigning the buzzing sound to the respective bee taxa.

user. Parts with no bee sound were not selected, but were kept for the subsequent analyses. We performed these analyses using the Raven Lite software (Cornell Laboratory of Ornithology, Ithaca, New York). The length of recordings ranged from five seconds to over one minute.

## Audio feature extraction

After the acoustic processing, audio feature extraction was applied to transform raw audio data into features that explicitly represent properties of the data and may be relevant for classification. This process is carried out through the MFCC [45], which is present in the *librosa* library [46]. As input, the algorithm takes an audio segment (flight or sonication), which goes through the following steps: pre-emphasis, framing, windowing, Discrete Fourier Transform (DFT), and filter bank (applying Discrete Cosine Transform—DCT), as described by [45] (see Fig 3).

The Discrete Fourier Transform (DFT) was applied in each frame and we calculate the spectrum; and subsequently compute the filter banks, which are formed by triangular filters, spaced according to the MEL frequency scale; then we obtained the log-energy output of each one of the MEL filters. Finally, the MFCC coefficients were obtained by applying the inverse transformation of the cosine (DCT) to the logarithm of the energy coefficients obtained in the previous step. The parameters applied to generate the MFCC coefficients were maintained by default, except for the minimum frequency of each audio segment and the number of features that was set to 40. This was necessary because MFCC cannot generate a larger number of features and the average duration of the segments was small, approximately 1.54 seconds.

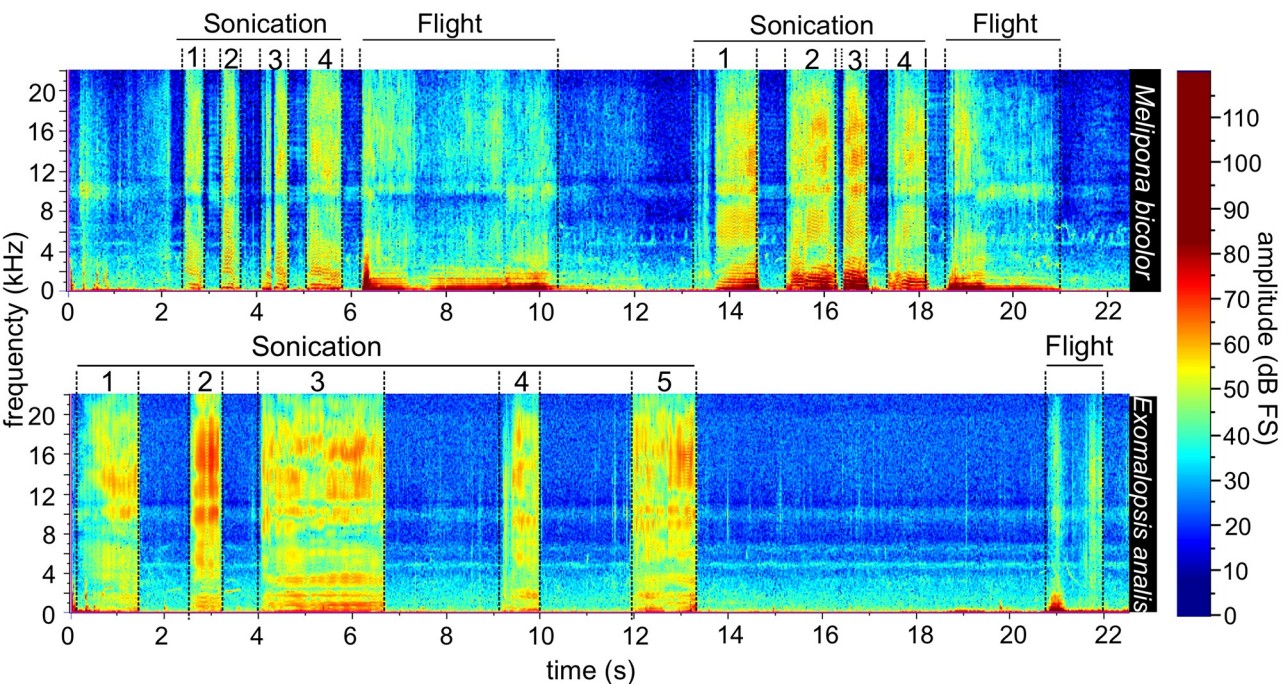

**Fig 2. Spectrograms of different types of buzzing (sonication and flight) for two visiting-bees species of tomato flowers (*Melipona bicolor* and *Exomalopsis analis*).** Note that the duration and amplitude and frequency of the buzzing-sounds vary between the species and among the type of buzzing.

**Similarity of the buzzing-sounds.** After audio feature extraction by the MFCC, we applied the euclidean distance score to estimate the similarity between the two types of sounds produced by bees visiting tomato flowers (sonication and flight). We calculated the euclidean distance score to all possible combinations between sonication and flight sounds and the average euclidean distance per bee taxon (species and genera). The euclidean distance has been used to measure sound similarities, especially in the human context (e.g. between the voice

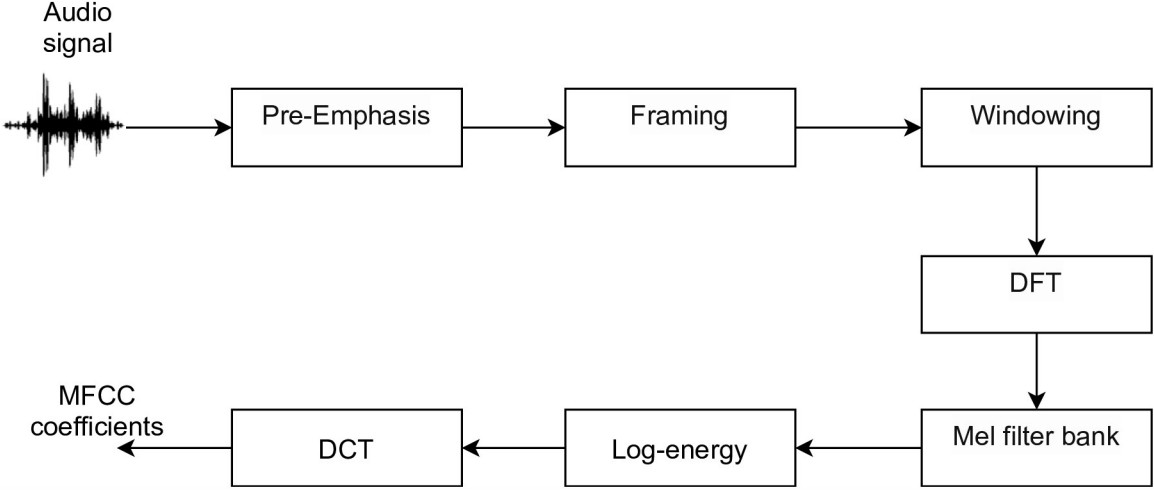

**Fig 3. Overview of the steps for audio feature extraction by Mel Frequency Cepstral Coefficients Method (MFCC)** Pre-emphasis, framing, windowing, Discrete Fourier Transform (DFT), and filter bank (applying Discrete Cosine Transform—DCT).

produced by different speakers [47], the speech of music and non-vocal sounds [48], and characteristics from the signals and to model its probability density functions [49]), being therefore, likely to be applied to bee buzzing classification. By definition, the greater the distance, the less similar the sonication and flight features would be. However, to simplify the interpretation of the results, we standardized the index between 0 and 1, where 1 represents maximum similarity and 0 no similarity (according to [50]).

## Classification

**Data splitting.**    During an exploratory analysis, we detected an unbalance of the sampled data between the classes (species/genera) and between the two behaviors (sonication and flight). There were 103 samples containing segments of flight and 218 of sonication. Moreover, the distribution of these segments by the classes was even more unbalanced (see Table 1). For example, the flight of the species *Augochloropsis brachycephala* was recorded just once. Considering that we need to distribute the data into training and testing sets, this species could not be part of both. This problem does not occur at the genus-level, because the number of classes decreases, consequently, the number of samples per class increases.

Due to the mentioned issues, the data division was stratified. As shown in Fig 1, the data set was divided into 50% for training and 50% for testing. The division was done through the function *train_test_split* of *scikit-learn* [51], this function is able to separate the data in a stratified way through the *StratifiedKFold* method. This method is a variation of *k-fold* that returns stratified folds: each set contains approximately the same percentage of samples of each target class as the complete set, dividing the classes by 50%.

After data division, it is necessary to apply standardization to the data set, this step is important for many machine learning estimators, because they can behave badly if the individual resources do not look more or less like standard normal distributed data set (for example, Gaussian with mean 0 and unit variance) [51]. Elements used in the objective function of a learning algorithm (such as the RBF kernel of Support Vector Machines or the $L_1$ and $L_2$ regularizers of linear models) assume that all resources are centered around 0 and have variance in the same order. If a feature has a variance that is orders of magnitude greater than others, it may dominate the objective function and make the estimator unable to learn from other features correctly as expected. Therefore, in order to solve this problem, data normalization was done with the *StandardScaler* method.

**Machine learning algorithms.**    Machine Learning techniques have demonstrated high efficiency and accuracy for classification of bumblebees and other groups of bees based on the characteristics of their buzzing-sounds [19, 35, 36]. Therefore, we chose some of the most common used ML classifiers to recognize the taxonomic identity of bees during visits to tomato flowers (according to S1 Table): Logistic Regression [52], Support Vector Machines [53, 54], Random Forest [55], Decision Trees [56, 57], and a classifier ensemble [58, 59]—a combination of multiple and diversified classifiers to generate a single classifier model. Ensemble methods train multiple learners to solve the same problem [59]. In contrast to classic learning approaches, which construct one learner from the training data, ensemble methods construct a set of learners and combine them. Therefore, we combine three classifiers (Random Forest, SVM e Logistic Regression) by majority vote. These classifiers were chosen because they achieved the best performance in recognizing bees buzzing-sounds (see S1 Table).

**Tuning the hyperparameters.**    In many cases, the performance of an algorithm in a given learning task depends on its hyperparameter settings. In order to obtain the best performance, the hyperparameters have been thoroughly tested. Methods to tune hyperparameters to the

problem at hand require the definition of a search space: the set of hyperparameters and ranges that need to be considered.

An important problem is to decide which hyperparameters should be considered, because a very large set of hyperparameters is computationally expensive and becomes more expensive as the search space augments. So far, there is no empirical evidence on which hyperparameters are most important to adjust and which hyperparameters result in similar performance when set to a reasonable default value. Hyperparameters that fall into this last category can be completely eliminated from the search space when the computational resources are limited [60].

We used the *GridSearchCV* method found in the *scikit-learn* library [51]. This method performs an exhaustive search, as input, it receives an estimator, a hyperparameter dictionary, and the cross-validation method then creates a model for each combination. Cross-validation is used to evaluate each individual model, this step divides the training data into 5 *folds*, see Fig 1. The 5 *folds* are also built with the *StratifiedKFold* method. After training and validation, the *GridSearchCV* method returns the model that achieved the best performance and uses it for the test set.

The sets of hyperparameters were defined as follows: for SVM, we vary the Kernel (Rbf, polynomial, sigmoid, linear), $C$ ranging in $\{0.001, 0.01, 0.1, 1, 10\}$, and $\gamma$ ranging in $\{1e-2, 1e-3, 1e-4\}$. For Logistic Regression, we considered penalty $\{l_1, l_2\}$ and $C$ ranging in $\{0.001, 0.01, 0.1, 1, 10\}$. For Decision Trees, we validated the "Gini impurity" and "entropy" for the information gain, these functions are used to measure the quality of a split in the tree. For Random Forest, we considered the number of trees in the forest varying in $\{100, 200\}$. Finally, for the ensemble model we vary only the $C$ parameter as already described, this is due to the great computational processing that this model requires.

## Evaluation metrics

To evaluate the performance of the classification generated by the algorithms and baselines, we used the following metrics: Accuracy (Acc), Macro-Precision (MacPrec), Macro-Recall (MacRec) and Macro-F1 (MacF1).

Let $i$ be a class from the set of classes $\mathcal{C}$. Let $\mathcal{T}$ be test set and let $c$ be a classifier, such that $c(t) = l$, where $t$ is an element of the test set $\mathcal{T}$ and $l \in \mathcal{C}$ is a *label* corresponding to a class in $\mathcal{C}$ assigned to $t$ by $c$. Let $g(t)$ be the ground truth class label of $t$. In regard to the $c$ classifier we define:

- *True Positives of class i*, denoted by $TP_i$, as the number of elements in $\mathcal{T}$ correctly labeled with class $i$ by $c$, i.e., $TP_i = |\{t \in \mathcal{T} \,|\, c(t) = g(t) = i\}|$.

- *False Positives of class i*, denoted by $FP_i$, as the number of elements in $\mathcal{T}$ that were wrongly classified by $c$ as belonging to class $i$. Formally, $FP_i = |\{t \in \mathcal{T} \,|\, c(t) = i \wedge g(t) \neq i\}|$.

- *False Negatives of class i*, denoted by $FN_i$, as the number o elements in $\mathcal{T}$ belonging to class $i$ but classified by $c$ with a label different from $i$, that is, $FN_i = |\{t \in \mathcal{T} \,|\, c(t) \neq i \wedge g(t) = i\}|$.

The above numbers are used to define traditional effectiveness measures of classifiers. These measures are: *Precision*, *Recall* and *F1* [61]. Precision $p(c, i)$ of a classifier $c$ in relation to a class $i$ is defined in Eq 1

$$p(c, i) = \frac{TP_i}{TP_i + FP_i}.$$  (1)

Informally, precision is the ratio between the number of test elements correctly labeled by $c$ with the class label $i$ and the number of all elements labeled (correctly or incorrectly) by $c$.

Recall, denoted by $r(c, i)$, of a classifier $c$ in relation to a class $i$ is defined by Eq 2

$$r(c, i) = \frac{TP_i}{TP_i + FN_i}. \tag{2}$$

Thus, recall is the ratio between the number of test elements belonging to class $i$ which were correctly labeled by $c$ and the total number of test elements of class $i$.

The $F_1$ measure is a combination of the precision and recall measures and is defined by Eq 3.

$$F1(c, i) = \frac{2p(c, i)r(c, i)}{p(c, i) + r(c, i)} \tag{3}$$

When comparing the effectiveness of classifiers generated from distinct learning methods, it is common to use a global measure of effectiveness. A global measure aims at resuming the effectiveness of the classifier over all classes in the test set. In this work we use the following global measures to compare the results of classifiers we use: *Accuracy* (*Acc*) (which is equivalent to *Micro-F1*), *Macro-Precision* (*MacPrec*), *Macro-Recall* (*MacRec*) and *Macro-F1* (*MacF1*). *Accuracy* of a classifier $c$ is the fraction of test elements that were correctly labeled by $c$, and is formally defined by Eq 4

$$Acc(c) = \frac{\sum_{i=1}^{|\mathcal{C}|} TP_i}{\sum_{i=1}^{|\mathcal{C}|}(TP_i + FP_i)} \tag{4}$$

The Macro measure (Macro-Precision, Macro-Recall and Macro-F1) is the average of the corresponding measure (Precision, Recall and F1) over all classes and are defined by Eqs 5, 6 and 7.

$$MacPrec(c) = \frac{\sum_{i=1}^{|\mathcal{C}|} p(c, i)}{|\mathcal{C}|} \tag{5}$$

$$MacRec(c) = \frac{\sum_{i=1}^{|\mathcal{C}|} r(c, i)}{|\mathcal{C}|} \tag{6}$$

$$MacF1(c) = \frac{\sum_{i=1}^{|\mathcal{C}|} F1(c, i)}{|\mathcal{C}|} \tag{7}$$

## Baselines establishment

To assess and compare the performance of ML algorithms in recognizing bees based on their buzzing sounds, we built three baselines. The first one, named "fundamental frequency" was estimated to compare our results —based on ML techniques and audio feature extraction by the MFCC—with results obtained by [43]—based on differences in the average fundamental frequency of the bees buzzing. The fundamental frequency was obtained by dividing each sound recording into three sections of similar duration and the lowest frequency of each section was measured using Avisoft-SASLab Lite (Avisoft Bioacoustics, Germany); the average fundamental frequency of each sound recording was the mean of the three frequencies, as performed by [43]. Then, the values of average fundamental frequency (±SD) were associated

with the corresponding bee taxon (species/genus). The species/genus whose average was between the lowest and highest frequency will be selected and the species/genus that has the lowest standard deviation will be predicted. For the second baseline, named as "Fundamental frequency (SVM)", we employed the best classifier here (based on the best F1-score) to recognize the bees taxa based only on the fundamental frequency. Lastly, we report the result of a majority baseline that assigns all the classes to the majority class, that is *Exomalopsis* for genus-level and *Exomalopsis analis* for species-level classification.

## Results

### Acoustic characteristics of the buzzing sounds

The acoustic proprieties (amplitude, frequency, and duration) of the buzzing can vary depending on the behavioral activity and bee species visiting tomato flowers. For example, the spectrograms of *Melipona bicolor* and *Exomalopsis analis* show that the flight and sonication buzzing-sounds are distinct from each other (Fig 2). During the flight (Fig 2), the spectrograms are time-independent and consist of a continuous frequency; amplitude variations may be related to the intensity of the sound over time, since the distance from the bee to the microphone can also vary. During the sonication, the fundamental frequency increases significantly (around 240 Hz) and the amplitude reaches higher values at higher frequencies.

The acoustic proprieties of the buzz are also different among the bee species. For example, while *M. bicolor* (Fig 2 upper spectrogram) presents successive short sonication buzzing-sounds, with brief breaks among them, the *E. analis* (Fig 2 bottom spectrogram) shows sonication intervals with irregular duration (generally longer than *M. bicolor*) and longer breaks among them.

### Performance of the machine learning algorithms

**Regarding the type of buzzing-sound.** The sonication and flight features extracted by the MFCC can be easily distinguished. They presented very low similarity among each other, ranging from 0.01 to 0.03 (Euclidean distance score) for bee species and between 0.01–0.02 for the genus. Moreover, the type of buzzing-sound also influenced the capacity of classifiers to recognize the visiting bees of tomato flowers. The ML algorithms reached a better performance, recognizing bees at species level based on sonication sounds (based on the best Macro-F1 score, see Table 2). The accuracy and Macro-F1-score were higher in classifications considering only the segments of floral sonication sounds rather than those of flight (Table 2). The sonication sounds classified by the SVM algorithm achieved the best performance among all combinations tested here (Accuracy = 73.39%; Macro-F1 = 59.06%, Table 2).

Nonetheless, at genus-level recognition, the performance of the algorithms did not seem to depend on the type of buzzing-sound (based on the higher Macro-F1 measure, Table 3). However, the buzzing sounds from flights led to a marginally better ML algorithms performance than sonication in recognizing the genera of bees (Table 3).

**Regarding the level of taxonomic resolution.** The performance of ML classifiers was different for acoustic recognition of bees at species and genus levels. Indeed, the complexity increases for species recognition in relation to genus recognition: there are 15 classes (against 8 genera) and the number of samples for some of them is very small ($N \leq 5$). The SVM reached the best Macro-F1 values at genus-level recognition (Flight, 60.2%; Table 3), which was similar to the Macro-F1 obtained by the SVM in species recognition (Sonication, 59.06%; Table 2). However, based on Accuracy, the Ensemble was the best for genus recognition and the SVM for species (Tables 3 and 2).

**Table 2. Predictive performance of different Machine-Learning algorithms on acoustic recognition of bee species based on the type of buzzing-sound (flight, sonication, and flight+sonication) during visits to tomato flowers.** The performance of the ML algorithms was measured by Accuracy (Acc), Macro-Precision (MacPred), Macro-Recall (MacRec) and Macro-F1 (MacF1) and compared with three baselines scenarios: (1) Majority class: assigning all the classes to the majority class; (2) Fundamental frequency: bees recognition based solely on the average frequency of the sonication, as performed by [43]; (3) Fundamental frequency (SVM): bees recognition based fundamental frequency and using the SVM algorithm, classifier with the best performance (based on the MacF1-score). Bold numbers represent the best results per evaluation metric within buzz-sound; Different upper side letters denote significant differences in the F1-score among the algorithms of the same buzzing-behavioral ($p \leq 0.05$, T-test); (**) denotes that the performance of the algorithm is higher than the baselines (based on the MacF1 measure; $p \leq 0.05$, T-test).

| Flight | | | |
|---|---|---|---|
| **Algorithms** | **Acc (%)** | **MacPrec (%)** | **MacRec (%)** | **MacF1 (%)** |
| LR** | 51.92 | 47.25 | 41.30 | 40.20[a] |
| SVM** | **55.76** | **56.91** | **53.78** | **49.00**[a] |
| RF | 48.07 | 47.32 | 43.20 | 41.46[a] |
| DTree | 26.92 | 24.25 | 23.88 | 19.74[b] |
| Ensemble** | 50.00 | 45.80 | 38.01 | 36.02[b] |
| **Sonication** | | | |
| **Algorithms** | **Acc (%)** | **MacPrec (%)** | **MacRec (%)** | **MacF1 (%)** |
| LR** | 64.22 | 45.56 | 41.13 | 41.27[a] |
| SVM** | **73.39** | **61.75** | **60.70** | **59.06**[b] |
| RF** | 58.71 | 47.73 | 34.72 | 37.67[a] |
| DTree | 43.11 | 31.31 | 35.35 | 29.07[c] |
| Ensemble** | 68.80 | 48.50 | 45.59 | 44.19[a] |
| **Flight + Sonication** | | | |
| **Algorithms** | **Acc (%)** | **MacPrec (%)** | **MacRec (%)** | **MacF1 (%)** |
| LR** | 53.41 | 53.67 | **51.36** | **48.61**[a] |
| SVM** | 56.52 | 46.66 | 46.59 | 45.16[a] |
| RF** | 50.31 | 44.56 | 38.13 | 36.21[b] |
| DTree | 32.91 | 33.02 | 28.19 | 25.91[c] |
| Ensemble** | **58.38** | **53.81** | 47.62 | 47.36[a] |
| **Baselines** | | | |
| | **Acc (%)** | **MacPrec (%)** | **MacRec (%)** | **MacF1 (%)** |
| Majority class | 23.00 | 2.00 | 7.00 | 2.00 |
| Fundamental frequency | 51.00 | 25.00 | 40.00 | 28.00 |
| Fundamental frequency (SVM) | 35.00 | 27.00 | 24.00 | 24.00 |

Just the LR and SVM classifiers always presented Macro-F1 values higher than the baselines at genus-level classification (S2 Table). On the other hand, only the DTree continually achieved lower performance than the baselines (based on the Macro-F1 measure, S2 Table). Considering the species recognition, besides the LR and SVM, the Ensemble also reached a better score than the baselines (based on the Macro-F1 measure, S3 Table).

The confusion matrix shows the number of correctly predicted genera versus erroneously predicted genera by SVM, the classifier with the best performance here (based on Mac-F1 score, Table 4). The SVM was capable to correctly recognize 64% (34 of 53) of the flight sounds samples. However, the capacity of the algorithm to identify bees was unequal among the genera. The SVM was able to correctly recognize more than 50% of the samples of four out of eight genera (*Bombus*, *Centris*, *Melipona* and *Eulaema*, Table 4).

On the other hand, for species-level recognition, the SVM was able to correctly predict 79% of the sonication samples (80 of 109) (Table 5). Moreover, this algorithm was capable to recognize *E. analis* (28 of 30 samples), the most representative species (Table 5). Moreover, this algorithm correctly recognized some species with a small number of samples, like

**Table 3. Predictive performance of different Machine-Learning algorithms on acoustic recognition of bee genera based on the type of buzzing-sound (flight, sonication, and flight+sonication) during visits to tomato flowers.** The performance of the ML algorithms was measured by Accuracy (Acc), Macro-Precision (MacPrec), Macro-Recall (MacRec) and Macro-F1 (MacF1) and compared with three baseline scenarios: (1) Majority class: assigning all the classes to the majority class; (2) Fundamental frequency: bee recognition based solely on the average frequency of the sonication, as performed by [43]; (3) Fundamental frequency (SVM): bee recognition based fundamental frequency and using the SVM algorithm, classifier with the best performance (based on the MacF1 score). Bold numbers represent the best results per evaluation metric within buzz-sound; Different upper side letters denote significant differences in the MacF1 scores among the algorithms of the same buzzing-behavioral ($p \leq 0.05$, T-test); (**) denotes that the performance of the algorithm is higher than the baselines (based on the MacF1 measure; $p \leq 0.05$, T-test).

| | Flight | | | |
|---|---|---|---|---|
| **Algorithms** | **Acc (%)** | **MacPrec (%)** | **MacRec (%)** | **MacF1 (%)** |
| LR** | 60.37 | **65.25** | 56.63 | 57.02[b] |
| SVM** | **64.15** | 64.44 | **60.49** | **60.20[a]** |
| RF | 54.71 | 45.85 | 41.22 | 38.17[c] |
| DTree | 39.62 | 20.92 | 28.79 | 21.85[c] |
| Ensemble | 60.37 | 64.37 | 56.84 | 55.23[a, b] |
| | **Sonication** | | | |
| **Algorithms** | **Acc (%)** | **MacPrec (%)** | **MacRec (%)** | **MacF1 (%)** |
| LR** | 60.90 | 56.72 | 49.91 | 51.55[b] |
| SVM** | 66.36 | **71.04** | **54.70** | **58.06[a]** |
| RF | 62.72 | 44.83 | 36.93 | 37.77[d] |
| DTree | 49.09 | 30.59 | 29.51 | 29.82[e] |
| Ensemble | **67.27** | 47.97 | 41.45 | 42.60[c] |
| | **Flight + Sonication** | | | |
| **Algorithms** | **Acc (%)** | **MacPrec (%)** | **MacRec (%)** | **MacF1 (%)** |
| LR** | 62.34 | 51.49 | 55.76 | 52.38[a] |
| SVM** | 67.90 | 57.79 | **60.46** | **58.59[a]** |
| RF** | 61.11 | 53.57 | 45.77 | 46.92[b] |
| DTree | 45.06 | 34.53 | 35.80 | 34.23[c] |
| Ensemble** | **68.61** | **61.23** | 56.99 | 58.09[a] |
| | **Baselines** | | | |
| **Baselines** | **Acc (%)** | **MacPrec (%)** | **MacRec (%)** | **MacF1 (%)** |
| Majority class | 30.00 | 4.00 | 12.00 | 6.00 |
| Fundamental frequency | 68.00 | 41.00 | 50.00 | 43.00 |
| Fundamental frequency (SVM) | 48.00 | 29.00 | 29.00 | 28.00 |

**Table 4. Confusion matrix with the best performance for bee buzzing-sounds classification at genus-level using MFCC features (flight with SVM classifier,** *MacF1* **= 60.20% and** *Acc* **= 64.15%).** The numbers in the matrix correspond to correctly (diagonal elements, bold) and incorrectly (out-of-diagonal elements) recognized samples in the data set. The best parameters of this classification were C = 10, decision_function_shape = "ovo", gamma = 0.01, kernel = "rbf".

| Predict → <br> True ↓ | *Augochloropsis* | *Bombus* | *Centris* | *Eulaema* | *Exomalopis* | *Melipona* | *Pseudaugochlora* | *Xylocopa* | All |
|---|---|---|---|---|---|---|---|---|---|
| *Augochloropsis* | **3** | 0 | 0 | 0 | 3 | 1 | 1 | 0 | 8 |
| *Bombus* | 0 | **10** | 0 | 0 | 0 | 0 | 0 | 0 | 10 |
| *Centris* | 0 | 0 | **5** | 0 | 0 | 2 | 0 | 0 | 7 |
| *Eulaema* | 0 | 0 | 0 | **1** | 0 | 0 | 0 | 0 | 1 |
| *Exomalopis* | 0 | 1 | 0 | 0 | **4** | 1 | 0 | 2 | 8 |
| *Melipona* | 0 | 1 | 1 | 0 | 2 | **7** | 0 | 1 | 12 |
| *Pseudaugochlora* | 0 | 0 | 0 | 0 | 1 | 0 | **0** | 0 | 1 |
| *Xylocopa* | 0 | 2 | 0 | 0 | 0 | 0 | 0 | **4** | 6 |

**Table 5. Confusion matrix with the best performance for bees buzzing-sounds classification at species-level using MFCC features (sonication with SVM classifier, *MacF1* = 59.06% and *Acc* = 73.39%).** The numbers in the matrix correspond to correctly (diagonal elements, bold) and incorrectly (out-of-diagonal elements) recognized samples in the data set. The best parameters of this classification were C = 10, decision_function_shape = "ovo", gamma = 0.01, kernel = "rbf".

| Predict → / True ↓ | A. brachycephala | Augochloropsis sp.1 | Augochloropsis sp.2 | B. morio | B. pauloensis | C. tarsata | C. trigonoides | E. nigrita | E. analis | E. minor | M. bicolor | M. quadrifasciata | P. graminea | X. nigrocincta | X. suspecta | All |
|---|---|---|---|---|---|---|---|---|---|---|---|---|---|---|---|---|
| A. brachycephala | **1** | 0 | 0 | 0 | 0 | 0 | 0 | 0 | 0 | 0 | 0 | 0 | 0 | 0 | 0 | 1 |
| Augochloropsis sp.1 | 0 | **2** | 0 | 0 | 1 | 0 | 2 | 0 | 0 | 0 | 0 | 0 | 0 | 0 | 1 | 6 |
| Augochloropsis sp.2 | 0 | 0 | **9** | 0 | 0 | 0 | 0 | 0 | 0 | 0 | 0 | 0 | 0 | 0 | 0 | 9 |
| B. morio | 0 | 0 | 0 | **5** | 1 | 0 | 1 | 0 | 0 | 0 | 0 | 0 | 0 | 1 | 0 | 8 |
| B. pauloensis | 0 | 0 | 0 | 0 | **6** | 0 | 0 | 0 | 0 | 0 | 1 | 0 | 0 | 1 | 0 | 8 |
| C. tarsata | 0 | 0 | 0 | 0 | 1 | **0** | 0 | 0 | 0 | 0 | 0 | 0 | 0 | 1 | 0 | 2 |
| C. trigonoides | 0 | 0 | 0 | 0 | 0 | 0 | **2** | 0 | 0 | 0 | 0 | 0 | 0 | 0 | 0 | 2 |
| E. nigrita | 0 | 0 | 0 | 0 | 0 | 0 | 0 | **0** | 1 | 0 | 0 | 0 | 0 | 0 | 0 | 1 |
| E. analis | 0 | 1 | 0 | 0 | 0 | 0 | 0 | 0 | **28** | 0 | 0 | 0 | 0 | 1 | 0 | 30 |
| E. minor | 0 | 0 | 0 | 0 | 0 | 1 | 1 | 0 | 2 | **7** | 0 | 0 | 0 | 0 | 0 | 11 |
| M. bicolor | 0 | 0 | 0 | 0 | 0 | 0 | 1 | 0 | 1 | 0 | **9** | 1 | 0 | 3 | 0 | 15 |
| M. quadrifasciata | 0 | 0 | 0 | 0 | 0 | 0 | 0 | 0 | 0 | 0 | 0 | **2** | 0 | 0 | 0 | 2 |
| P. graminea | 0 | 0 | 0 | 0 | 0 | 0 | 1 | 0 | 1 | 0 | 1 | 0 | **8** | 0 | 0 | 11 |
| X. nigrocincta | 0 | 0 | 0 | 0 | 0 | 0 | 0 | 0 | 0 | 0 | 0 | 0 | 1 | **1** | 0 | 2 |
| X. suspecta | 0 | 0 | 0 | 0 | 0 | 0 | 0 | 0 | 0 | 0 | 0 | 0 | 0 | 1 | **0** | 1 |

*Augochloropsis brachycephala*, *Augochloropsis* sp.2, *Melipona quadrisfaciata* and *Centris trigonoides*.

## Discussion

The accuracy of tested ML algorithms in recognizing flower-visiting bees of tomato crops ranged from 49 to 74% on a data set of 59 audio recording samples. The algorithms reached a better performance to assign the bees buzzing sounds to their respective taxa than frequency-based trials. Moreover, we found that the sonication sounds are more relevant to bees species recognition. The ML algorithms achieved a greater performance in recognizing bee species when we considered only the sounds produced during sonication. On the other hand, the genera recognition was not dependent on the type of buzzing-sound.

### Advantages of machine-learning over classifications based on fundamental frequency

The ML algorithms achieved higher performance recognizing bee taxa than analyses based on fundamental frequency and realized on the same data set. Moreover, the statistical analysis based on fundamental frequency differences performed by [43], failed to distinguish between most bee species. In fact, analyses based solely on the fundamental frequency (average frequency) must lose part of the intrinsic complexity of buzzing sounds, which is multifactorial and time-dependent [12]. The buzz has other acoustic features, like the amplitude and duration, that combined between them and with the frequency must contribute to characterize the buzzing-sounds [62]. However, this must result in a huge amount of data with unusual distributions, non-linearity, complex data interactions, dependence on the observations that would not be well handled by commonly used statistical methods in ecology [63, 64]. On the other hand, the ML algorithms combined with the MFCC method has been able to correctly predict 66% of all samples; 79% of the samples of species based on sonication sounds and SVM algorithm. Likely due to the ML attributes boosted by MFCC, we reached here a higher performance on acoustic recognition of bees than classifications based only on the fundamental frequency.

### The recognition of bees depends on the type of buzzing

There are pronounced differences between the biomechanical properties of the buzzing produced during sonication and those produced during the flight [38]. The sonication sounds have amplitudes and frequencies higher than flight buzzes [38]. The flight sound has few oscillations and roughly time-independent. It consists of the natural frequency (the frequency at which the wings oscillate) and its higher harmonics [37, 38]. Therefore, the flight buzzing can be more similar among species over the higher-level taxa. This may be the reason why flight sounds were more relevant to the recognition of the genus than to the species. Besides that, the incorporation of both buzzing sounds (sonication+flight) does not seem to interfere with the performance of the algorithms in recognizing bees at genus-level, because the performance of the ML algorithms was similar.

On the other hand, the sonication sounds were associated with higher performance in recognizing bee species. Although, the mechanical characteristics of the sonication have been related to the amount of pollen released from poricidal anthers [12, 43, 65–67], the acoustic properties of buzzing-sounds are also related to intrinsic attributes of the species [39–42]. Consequently, the higher specificity related to sonication sounds makes it more relevant to species recognition by the ML algorithms. Although the behavioral context that the buzzing-sound

was produced was not relevant for genera recognition by ML algorithms, it was for the recognition of species.

## Limitations of buzzing-sound classification with machine learning

Although here we classified a greater taxonomic diversity of flower-visiting bees based on their buzzing than previous studies (see S1 Table); grouping the largest number of genera and families of bees (15 species from 8 genera and 2 subfamilies; Table 1), the machine-learning approach presented some limitations in recognizing bees based on buzzing sounds. Firstly, the ML algorithms are domain-dependent. This means that a classifier can perform very well when it is applied on the same domain to the one it was trained, yet the performance decreases when it is applied to a different domain (e.g. species/genus, sonication/flight). Thus, the classifier needs to be retrained in order to perform well on a different domain. Secondly, the performance of ML algorithms was not homogeneous among the classes of species and genera. The performance was very high for some bee taxa, especially the most sampled, and varied for unrepresentative taxa (*Augochloropsis* sp.2, 100%; *E. analis*, 93%; *P. graminea*, 72%), which may be related to the unbalanced number of samples per bee taxa, an issue also reported by related studies (see S1 Table). Consequently, bees that rarely visited the tomato flowers and/or were difficult to capture were under-sampled. This bias is inherent to the system since the local abundance of individuals per species naturally varies and the bees spontaneously visit the flowers. On the other hand, to ensure taxonomic identification, all specimens had to be captured. This was a requirement to include the buzz sound associated with a given bee on the acoustic analysis, in case the bee could not be sampled, we deleted the corresponding sound file. The oil collecting bees (*Centris* sp.), for example, were more difficult to sample, because they flew quite fast between flowers, remained for a short time in the same flower, and/or did not visit nearby flowers, which make it difficult to follow them. The general consequence of these two factors mentioned above (different abundance of individuals among bee taxa and sampling bias) was an unbalanced sampling among classes of buzzing bees. Unfortunately, the Machine Learning algorithms have a considerable loss of performance in classifying unbalanced data [68].

Nevertheless, the performance in recognizing the buzzing sounds was uneven among the ML algorithms. The LR and especially the SVM outstanding from the other algorithms and constantly obtained better performance than the baselines. The SVM through weighted evaluation metrics stood out among the classifiers, achieving the best performance in recognizing the visiting-bees of tomato. In fact, the SVM has strong theoretical foundations with excellent empirical successes [69] and has demonstrated tolerance to data sets with few samples per class and unbalanced data (see Evaluation metrics in text) [70]. This may be the main reason that this classifier was almost always produced the best classifications in relation to baselines and other algorithms. Despite that, the SVM performance is still lowed on the small data set tested here, compared to ML standards (see S1 Table). Therefore, we suppose that there is so little audio data that no classical ML classifier can, in principle, generalize well on it. Further studies, considering larger recording samples, and/or applying algorithms that can perform more complex processing tasks like unsupervised learning systems (e.g., clustering, dimensionality reduction, recommender systems, deep learning) may reach better classification performance.

## Consequences of automating the bee recognition to tomato yields

The bee identity is associated with pollination effectiveness and fruit yields since the performance as pollinators tends to be different among species/groups of visiting bees (e.g. [4, 6–9,

16]. Differences in the body size of the bees in relation to the distance between anthers and stigma may be a key factor in explaining this. Larger pollinators transfer more pollen than smaller ones [71], since their body size fits or exceeds the distance between anthers and stigma [72–74]. Therefore, automating the taxonomic recognition of flower-visiting bees would be especially relevant for tomato production, whereas the quality of the pollination provided is linked to the identity of the bee. Then, farmers, agronomists, and other professionals interested in improving the pollination of cultivated tomatoes could identify the species of visiting bees without needing an expert in insect taxonomy. Aware of the value of bees to the crop income, the farmer could be motivated to adopt practices to benefit the most successful pollinators and indirectly the overall local bee community, promoting profitable and sustainable agricultural practices.

Moreover, the automated taxonomic recognition of bees may apply to other buzz-pollinated plants, since they are primarily visited by bees that produce buzzing sounds to extract pollen [9, 12, 14, 15]. Some of these plants are, as well as tomato, important food crops, like blueberry, kiwi, cranberry, and eggplant [9, 75–77]. However, some procedures must be adopted to avoid sampling bias and facilitate acoustic recognition: (1) studies must focus on one plant species because the same bee species produce vibrations with different frequencies and duration when visiting different plant taxa [42, 78]; (2) consider the limitations when analyzing the relative acoustical amplitude because this energy-related parameter is dependent on measurement procedures (e.g. the recorder model and configuration, the distance of the focal object) and does not necessarily correspond to the vibrational amplitude [19, 79].

In summary, the ML algorithms powered by the MFCC feature extraction method could lead to automate the taxonomic recognition of flower-visiting bees of tomato crop. We found advantages of ML classifiers in recognizing species of bees based on their buzzing sounds over conventional analyzes based on fundamental frequency alone [43]. Some classifiers, especially the SVM, an algorithm that better handles a data set of low sampling, achieved better performance in relation to the randomized and sound frequency-based trials. The buzzing sounds produced during sonication were more relevant for the taxonomic recognition of bees species than the flight sounds. On the other hand, we found that the ML classifiers achieve better performance to recognize bee genera based on flight sounds. As far as we know, the use of ML algorithms to explore these two kinds of bee sounds for bee taxa identification has not been reported previously. Future studies may focus on the extension of this approach to other buzz-pollinated crops as well as on the technological application of this model, for example, the development of apps based on ML techniques and compatible with smartphones.

## Supporting information

**S1 Table. Overview of the studies applying machine learning and audio feature extraction methods to the acoustic monitoring/detection of bees.**
(PDF)

**S2 Table. Pairwise comparison of the performance of the machine-learning algorithms and baseline scenarios (majority class, fundamental frequency, and fundamental frequency (SVM)) in acoustic recognition of bee genera based on buzzing-sounds produced during three behavioral contexts (flight, sonication, and flight + sonication).** Internal numbers correspond to P-values obtained by the T-test; P-values highlighted in bold ($p \leq 0.05$) indicate significant differences among the F1-score of the ML algorithms/baselines.
(PDF)

**S3 Table. Pairwise comparison of the performance of the machine-learning algorithms and baseline scenarios (majority class, fundamental frequency, and fundamental frequency (SVM)) in acoustic recognition of bee species based on buzzing-sounds produced during three behavioral contexts (flight, sonication, and flight + sonication).** Internal numbers correspond to P-values obtained by the T-test; P-values highlighted in bold ($p \leq 0.05$) indicate significant differences among the F1-score of the ML algorithms/baselines. (PDF)

## Acknowledgments

We thank Eduardo Almeida of the University of São Paulo (USP) and Fernando Silveira of the Federal University of Minas Gerais (UFMG) for identifying bees.

## Author Contributions

**Conceptualization:** Nádia Felix Felipe da Silva, Fernanda Neiva Mesquita, Thierson Couto Rosa, José Neiva Mesquita-Neto.

**Data curation:** Priscila de Cássia Souza Araújo.

**Formal analysis:** Alison Pereira Ribeiro, Nádia Felix Felipe da Silva.

**Funding acquisition:** José Neiva Mesquita-Neto.

**Investigation:** Alison Pereira Ribeiro, Fernanda Neiva Mesquita, Priscila de Cássia Souza Araújo, Thierson Couto Rosa, José Neiva Mesquita-Neto.

**Methodology:** Alison Pereira Ribeiro, Nádia Felix Felipe da Silva, Thierson Couto Rosa, José Neiva Mesquita-Neto.

**Project administration:** José Neiva Mesquita-Neto.

**Resources:** Priscila de Cássia Souza Araújo, José Neiva Mesquita-Neto.

**Software:** Alison Pereira Ribeiro.

**Supervision:** Nádia Felix Felipe da Silva, Thierson Couto Rosa, José Neiva Mesquita-Neto.

**Validation:** Alison Pereira Ribeiro, Priscila de Cássia Souza Araújo.

**Visualization:** Alison Pereira Ribeiro, Nádia Felix Felipe da Silva.

**Writing – original draft:** Alison Pereira Ribeiro, Nádia Felix Felipe da Silva, Fernanda Neiva Mesquita, José Neiva Mesquita-Neto.

**Writing – review & editing:** Nádia Felix Felipe da Silva, Fernanda Neiva Mesquita, Priscila de Cássia Souza Araújo, Thierson Couto Rosa, José Neiva Mesquita-Neto.

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
