## [Decision Letter · Decision Letter 0]

22 Jul 2021

Dear Dr. Mesquita-Neto,

Thank you very much for submitting your manuscript "Machine Learning approach for automatic recognition of tomato-pollinating bees based on their buzzing-sounds" for consideration at PLOS Computational Biology.

As with all papers reviewed by the journal, your manuscript was reviewed by members of the editorial board and by several independent reviewers. In light of the reviews (below this email), we would like to invite the resubmission of a significantly-revised version that takes into account the reviewers' comments.

The reviewers agree on the strengths of the article, but all have important feedback. In particular, I concur with many of the issues that Reviewer 2 raises. Thus we would require major corrections before we consider publishing the paper with Plos Computational Biology.

A definite issue is data availability: in the submission information, you state that the data are available from https://doi.org/10.1111/1744-7917.12602 but this links to a 2018 journal article which does not seem to offer any raw data. Plos Comp Biol has specific requirements for data availability (see https://journals.plos.org/ploscompbiol/s/submission-guidelines for information). If the data availability issue is not fixed, the article will certainly be rejected.

The same is true of software code: please see https://journals.plos.org/ploscompbiol/s/code-availability

As Reviewer 2 points out, the application of MFCCs in ML classifiers such as SVM, random forests, or decision trees to classify audio bee samples is not novel. It has been done before, and is now part of the peer-reviewed electronic beehive monitoring literature. (Lines 112-126 are redundant, and MFCCs can be cited to a standard textbook.) A resubmission should more clearly cite and review previous literature that uses MFCCs and machine learning for bee monitoring, even if the monitoring is of hives rather than pollination.

The abstract uses some non-standard English, so I also suggest some small improvements to the abstract:

"The bee-mediated pollination" -> "Bee-mediated pollination"

"it becomes primordial" -> "it is important"

"The ML techniques could lead to automate" -> "ML techniques could be used to automate"

I have highlighted specific issues here in my editor's decision, but all of the reviewers' detailed comments should be addressed.

We cannot make any decision about publication until we have seen the revised manuscript and your response to the reviewers' comments. Your revised manuscript is also likely to be sent to reviewers for further evaluation.

Sincerely,

Dan Stowell

Associate Editor

PLOS Computational Biology

Natalia Komarova

Deputy Editor

PLOS Computational Biology

Thank you for submitting your paper. The reviewers agree on the strengths of the article, but all have important feedback. In particular, I concur with many of the issues that Reviewer 2 raises. Thus we would require major corrections before we consider publishing the paper with Plos Computational Biology.

A definite issue is data availability: in the submission information, you state that the data are available from https://doi.org/10.1111/1744-7917.12602 but this links to a 2018 journal article which does not seem to offer any raw data. Plos Comp Biol has specific requirements for data availability (see https://journals.plos.org/ploscompbiol/s/submission-guidelines for information). If the data availability issue is not fixed, the article will certainly be rejected.

The same is true of software code: please see https://journals.plos.org/ploscompbiol/s/code-availability

As Reviewer 2 points out, the application of MFCCs in ML classifiers such as SVM, random forests, or decision trees to classify audio bee samples is not novel. It has been done before, and is now part of the peer-reviewed electronic beehive monitoring literature. (Lines 112-126 are redundant, and MFCCs can be cited to a standard textbook.) A resubmission should more clearly cite and review previous literature that uses MFCCs and machine learning for bee monitoring, even if the monitoring is of hives rather than pollination.

The abstract uses some non-standard English, so I also suggest some small improvements to the abstract:

"The bee-mediated pollination" -> "Bee-mediated pollination"

"it becomes primordial" -> "it is important"

"The ML techniques could lead to automate" -> "ML techniques could be used to automate"

I have highlighted specific issues here in my editor's decision, but all of the reviewers' detailed comments should be addressed.

Reviewer's Responses to Questions

**Comments to the Authors:**

Reviewer #1: In the paper, Pereira Ribeiro et al. studied the use of machine-learning algorithms to classify bees visiting the tomato flowers. They first collected a dataset containing flight and sonication buzzes of 15 bee species of 8 genera. After isolating the buzzing sounds, they developed a series of ML classification algorithms to work either on a species or on a genus level. They discuss the results in view of advantages and limitations and possibilities for future implementation in an automated tool for crop monitoring.

The paper is clearly written, the methodology is correct, and the results are reasonable, especially in view of the classification metrics of some related studies that the paper cites. The authors are well-aware of the issues related to the unbalanced data samples and I believe they handled it appropriately. In my opinion, the paper represents a substantial contribution to the field of bioacoustics in pollinator science and has promising implications for future studies.

I have some minor comments that the authors may consider.

• Introduction: Perhaps mention that the pollination of tomatoes in greenhouses is typically done by bumblebees that are reared particularly for the purpose.

• At some points, the reference is shown as (??), for example lines 61, 183, 191, etc.

• Acoustic pre-processing. Here, perhaps mention that the flight and sonication buzzes are pronouncedly different in structure, so they can be easily distinguished from the recordings afterwards (as clearly seen in one of the later figures).

• Line 294: “quite different” … consider using more concise term

• Early in the paper you mention that you would compare the ML results with those of conventional statistical methods, but you only cite a previous paper on the same dataset regarding statistical methods. I suggest you mention this the first time you talk about the comparison and write a couple of sentences more of the discussion.

• As a possible experiment, perhaps it would be interested to construct a binary classifier between the most common species and “others”, if the most common species is so much more abundant. But probably this is beyond the scope of the present paper.

Reviewer #2: Overall, I have enjoyed reading this article but it has serious problems that must be addressed before it is published.

Summary: The objective of the paper of the paper is to provide a preliminary demonstration that some ML algorithms can be used in recognizing the taxonomic identify of bees visiting tomato flowers from audio samples of their buzzing sounds.

The authors formulate two hypothesis:

Hypothesis 1: "Due to the high efficiency and accuracy demonstrated by ML tools in automatic sound classification, we expected that these ML algorithms would obtain a greater performance compared to conventional statistics."

Hypothesis 2: "Based on this, we predicted that the buzzing-sounds produced by floral vibrations are more relevant for recognizing bees taxa."

My Comments:

Comment 1: It was unclear to me as I was reading the paper what the authors mean by "conventional statistics." In the section "Advantages of Machine-Learning over conventional statistics" they give references to publication [31] and where they state that their analysis "based on fundamental frequency differences, failed to distinguish between most bee species."

Fundamental frequency differences are not conventional statistics. Perhaps, the authors could explain more clearly what they mean by "conventional statistics."

Comment 2: Line 61: Reference is missing in the sentence "Random Forest, Support Vector Machines, and Logistic Regression are the most applied classifiers, and Mel Frequency Cepstral Coefficients is the most used feature extraction strategy (see ??)."

Comment 3: Lines 91-92: "The acoustic recording of buzzes was carried out using tomato plants (Lycopersicon esculentum ac. BGH 7488) grown at the experimental fields of the Federal University of Viçosa (Minas Gerais, Brazil). A professional

hand recorder (SongMeter SM2, Wildlife Acoustics, USA) was used to record the buzzing sounds of bees visiting the tomato flowers."

This is a very insufficient description for me to assess the accuracy of the obtained samples. How were the audio samples obtained? Where was the recorder placed? Was it hand-held? How closely was it placed to the pollinating bees? For a method to be scientific, it must be replicated by other researchers. As a researcher I cannot replicate the data acquisition method of the authors. The authors should provide a picture/drawing of how they recorder was placed with respect to the bee and the tomato plans.

Comment 4: I might have missed it, but I did not see any references to the availability of the authors' dataset, which has a negative impact on replicability.

Comment 5: It looks like there were a total of 321 individual bees captured. Are these the bees from which the audio samples were obtained? How many audio files (wav files) were obtained from these bees?

Comment 6: Line 104: "Parts with no bee sound or high background noise disturbance were not considered."

This makes the results significantly weaker. This fact should be explicitly mentioned in the abstract. When an audio classifier is deployed in the real world, it must deal with the problem of noise, especially ambient noise such as lawn mowers, cars, human

speech, wind, music, etc.

Comment 7: Lines 106-107: "The length of individual recordings ranged from five seconds to over one minute."

How were these recordings distributed among the training and testing data?

Comment 8: The authors write: "After, the resulting data set was split into 50% for the training/development

set (delimited by the red dashed line) and 50% for the testing data set."

A more standard split is 70-30? Why 50-50? I guess there is an explanation of sorts in lines 161-166.

Comment 9: Lines 112 -- 126 are redundant. MFCCs are very well know since the 1980's in

the audio processing communities. A couple of references are sufficient.

Comment 10: Lines 134 - 136: "This was necessary because MFCC cannot generate a larger number of

features and the average duration of the fragments was small, approximately 1.54 seconds."

This is where I got very confused. In lines 106-107, the authors write: "The length of individual recordings ranged from five seconds to over one minute."

What is the difference b/w an individual recording and a fragment? What do

the authors mean by a "fragment"?

Comment 11: Lines 152-155: "During an exploratory analysis, we detected an unbalance of the sampled data between

the classes (species/genera) and between the two behaviors (sonication and flight). There were 103 sound samples referring to flight and 218 samples corresponding to sonication, totaling 321 samples."

So, if my understanding is accurate, 1) there are 321 individual bees (Table 1); 2) there are "fragments" with an average duration of approximately 1.54 seconds (lines 134-136); 3) there are a total 642 audio samples (lines 152-155).

How are these entities related? How are the 642 audio samples obtained? The authors must include another table with a detailed distribution of the samples among the bee tax: number of Augochloropsis brachycephala samples; number of Augochloropsis sp.1 samples; number of Augochloropsis sp.2 samples, etc.

Comment 12: Lines 188-191: "Therefore, we combine three classifiers (Random Forest, SVM e Logistic Regression)

by majority vote. These classifiers were chosen because they achieved the best performance in recognizing bees buzzing-sounds (see ??)."

There is a missing reference in the above sentence.

Comment 13: The authors report that the best flight perfromance metrics of the SVM classifier in Table 2 are 55.76, 56.91, 53.78, 49.00. These are weak by ML standards on such a small dataset. The sonification metrics of the SVM are better: 73.39 61.75 60.70 59.06, but are still very low, which suggests to me one of the three things:

1) the SVM, although the best among the reported classifiers, is not a good fit for this domain;

2) the MFCC features are either inappropriate or insufficient for this domain;

3) there is so little audio data that no classifier can, in principle, generalize well

on it.

I came to the same conclusions when going through the data in Table 3. The authors should

address these points both in the abstract and the discussion.

Comment 14: Lines 317-318: "Just the LR and SVM classifiers always presented Macro-F1 values higher than the

baselines at genus-level classification (??)."

There is a missing reference in the above sentence.

Comment 15: Lines 319-321: there are two missing references.

Comment 16: Lines 336-337: "The ML algorithms were capable of recognizing most of the flower-visiting bees of the

tomato crop, based only on the characteristics of their buzzing sounds."

This is a very vague statement, and a misleading one. A more accurate description would be that the tested ML algorithms' accuracy ranged from 49 to 74% on a dataset of 642 audio samples with the audio samples with ambient noise removed from the dataset.

Comment 17: Lines 358-362: "Moreover, the ML combined with the MFCC method has been able to correctly predict

66% of all samples; 79% of the samples of species based on sonication sounds and SVM algorithm. Likely due to the ML attributes and boosted by MFCC, we reached here higher performance on acoustic recognition of bees than the conventional statistics, thus, corroborating our hypothesis 1."

Comment 18: The authors never made clear what they mean by "conventional statistics." What exactly is being

compared? The comparisons from [31] should be briefly summarized in this article? What were they? Are they significantly better than the results reported in this article? Otherwise, there is little evidence corroborating hypothesis 1.

Comment 19: Many references are missing in the section "Advantages and limitations of buzzig-sound classifications

with Machine Learning."

Comment 20: Lines 381-382: "Therefore, corroborating our hypothesis 2 only for species recognition."

There is insufficient evidence to corroborate this hypothesis. This sentence should be either reworded or removed from the article altogether.

Reviewer #3: The manuscript ID PCOMPBIOL-D-21-01039 entitled "Machine Learning approach for automatic recognition of tomato-pollinating bees based on their buzzing-sounds" is an interesting and novel contribution about taxa recognition of bees which buzz-pollinated the tomato using buzzing sounds. However, it is not very clear why it is importance that farmers or agronomists know the taxonomical identification of buzzing bees. All buzzing bee could be consider efficient pollinators of tomato? In other solanaceous plants if bees is too small only function as thief or if is too big could damage the flower. Authors need to review more literature to justify better the use of this application. Does this ML is possible to use with other buzz-pollinated plants? It will be interesting to discuss this possibility. This application could be very useful to buzzing bee identification in other buzz-pollinated plants.

Also, I included the following minor’s corrections:

L17 Please review the actual scientific name of tomato. It seems that the actual is Solanum lycopersicum L. and the synonym is Lycopersicon esculentum Mill.

Please review “Augochloropsis” spelling in Table 4 and along the paper.

L61, 183, 191, 318, 320, 321, 395, 418. Please check “??”

**Have the authors made all data and (if applicable) computational code underlying the findings in their manuscript fully available?**

Reviewer #1: None

Reviewer #2: None

Reviewer #3: Yes

PLOS authors have the option to publish the peer review history of their article (what does this mean?). If published, this will include your full peer review and any attached files.

Reviewer #1: No

Reviewer #2: No

Reviewer #3: No
---

## [Decision Letter · Decision Letter 1]

6 Sep 2021

Dear Dr. Mesquita-Neto,

We are pleased to inform you that your manuscript 'Machine Learning approach for automatic recognition of tomato-pollinating bees based on their buzzing-sounds' has been provisionally accepted for publication in PLOS Computational Biology.

Best regards,

Dan Stowell

Associate Editor

PLOS Computational Biology

Natalia Komarova

Deputy Editor

PLOS Computational Biology

Reviewer's Responses to Questions

**Comments to the Authors:**

Reviewer #1: The authors have addressed my comments, together with the comments from the other two reviewers, the paper has been significantly improved. I now fully support the publication of the paper in the journal.

Reviewer #2: I've carefully read your responses to all reviewers' comments. I appreciate you taking time to write detailed responses and address all the issues in the reviewers' comments and making your dataset available. Thank you very much! I wish you best of luck with your research. This is an important and valuable research venue and I hope you'll continue to make valuable contributions to it.

Reviewer #3: The new version of manuscript entitled "Machine Learning approach for automatic recognition of tomato-pollinating bees based on their buzzing-sounds" is stronger than old version, and authors address majority of comments and suggestions for three reviewers. Authors explain deeper the methods than early version. In addition, they discussed the relevance of their results to other buzz-pollinated plants, and the applicability to tomato farmers.

I suggest only minor corrections:

L17. Incomplete parentheses

L107. I am not sure if the number and unit is join or separate. Please check!

**Have the authors made all data and (if applicable) computational code underlying the findings in their manuscript fully available?**

Reviewer #1: None

Reviewer #2: Yes

Reviewer #3: Yes

PLOS authors have the option to publish the peer review history of their article (what does this mean?). If published, this will include your full peer review and any attached files.

Reviewer #1: No

Reviewer #2: No

Reviewer #3: **Yes: **Lislie Solís-Montero

---

## [Editor Report · Acceptance letter]

10 Sep 2021

PCOMPBIOL-D-21-01039R1 

Machine Learning approach for automatic recognition of tomato-pollinating bees based on their buzzing-sounds

Dear Dr Mesquita-Neto,

I am pleased to inform you that your manuscript has been formally accepted for publication in PLOS Computational Biology. Your manuscript is now with our production department and you will be notified of the publication date in due course.

With kind regards,

Andrea Szabo
